# COVID-19 Presentation and Outcomes among Cancer Patients: A Matched Case-Control Study

**DOI:** 10.3390/cancers13215283

**Published:** 2021-10-21

**Authors:** Julien Péron, Tristan Dagonneau, Anne Conrad, Fanny Pineau, Sara Calattini, Gilles Freyer, David Perol, Christophe Sajous, Maël Heiblig

**Affiliations:** 1Medical Oncology Department, Institut de Cancérologie des Hospices Civils de Lyon, 69002 Lyon, France; gilles.freyer@univ-lyon1.fr (G.F.); christophe.sajous@chu-lyon.fr (C.S.); 2Laboratoire de Biométrie et Biologie Evolutive, Equipe Biostatistique-Santé, CNRS UMR 5558, Université Claude Bernard Lyon 1, 69100 Villeurbanne, France; 3Faculté de Médecine et de Maïeutique Lyon-Sud-Charles Mérieux, Université Claude Bernard Lyon 1, 69310 Pierre-Bénite, France; mael.heiblig@live.fr; 4Medical Information Department, Hospices Civils de Lyon, 69002 Lyon, France; tristan.dagonneau@chu-lyon.fr; 5Infectious Diseases Department, Croix-Rousse Hospital, 69004 Lyon, France; anne.conrad@chu-lyon.fr; 6Centre International de Recherche en Infectiologie (CIRI), Inserm U1111, Université Claude Bernard Lyon 1, CNRS, UMR5308, Ecole Normale Supérieure de Lyon, University of Lyon, 69007 Lyon, France; 7Clinical Research Plateform, Institut de Cancérologie des Hospices Civils de Lyon, 69002 Lyon, France; fanny.pineau@chu-lyon.fr (F.P.); sara.calattini@chu-lyon.fr (S.C.); 8Centre Léon Bérard, 69008 Lyon, France; david.perol@lyon.unicancer.fr; 9Haematology Department, Institut de Cancérologie des Hospices Civils de Lyon, 69310 Pierre-Bénite, France

**Keywords:** COVID-19, cancer, matched cohort study, mortality, prognosis

## Abstract

**Simple Summary:**

Cancer patients have been described in previous studies as having a higher risk of contracting COVID-19 and a higher risk of developing a severe form of the disease. In this study, we compared a group of patients hospitalized for COVID-19 within the Lyon area with a matched group of patients free of cancer and also hospitalized for COVID-19. Cancer patients and control patients were matched in order to have a similar age, gender, and other preexisting conditions. In this study, the risk of death was higher among cancer patients, but the intensity of care was lower (lower access to intensive care unit, shorter length of stay). The risk of death among cancer patients appears to be mainly driven by the severity of the infection and therapeutic limitations decided at admission.

**Abstract:**

It has been suggested that cancer patients are at higher risk of contracting COVID-19 and at higher risk of developing a severe form of the disease and fatality. This study’s objectives were to measure the excess risk of mortality and morbidity of patients with cancer among patients hospitalized for a SARS-CoV-2 infection, and to identify factors associated with the risk of death and morbidity among cancer patients. All first cancer patients hospitalized for COVID-19 in the two main hospitals of the Lyon area were included. These patients were matched based on age, gender, and comorbidities with non-cancer control patients. A total of 108 cancer patients and 193 control patients were included. The severity at admission and the symptoms were similar between the two groups. The risk of early death was higher among cancer patients, while the risk of intubation, number of days with oxygen, length of stay in ICU, and length of hospital stay were reduced. The main factors associated with early death among cancer patients was the severity of COVID-19 and the number of previous chemotherapy lines. The outcomes appear to be driven by the severity of the infection and therapeutic limitations decided at admission.

## 1. Introduction

The identification of patients at risk of developing a severe form of COVID-19 has been a matter of intensive research since the event of the COVID-19 pandemic in December 2019. Very early reports from China described that cancer patients were more likely to deteriorate into severe COVID-19 illness [1]. Patients with cancer have then been iteratively reported to be at increased risk of infection with SARS-CoV-2 and a more severe disease course [2,3,4].

However, cancer encompasses many different diseases and treatments, affecting a heterogeneous group of patients of all ages and with heterogeneous comorbidities. It has been described that patients with hematological cancer have a significantly increased case–fatality rate compared to patients with solid tumors [5]. The relationship between anticancer treatment and susceptibility to COVID-19 remains uncertain, but the mortality from COVID-19 in cancer patients has been reported to be mostly driven by age, gender, and comorbidities [6].

Overall, the causal relationship between cancer and the COVID-19 fatality rate remains poorly explained. Cancer patients are more likely to be exposed to transmissible agents due to their frequent contact with the health care system, cancer is likely to be associated with other possibly confounding comorbidities, limitation of therapeutic effort is more likely to occur among cancer patients, and some cancer treatments or cancer types might impair the quality of the patients’ immune response to SARS-CoV-2 infection and to vaccines [7,8].

This study’s objectives were to measure the excess risk of mortality and morbidity of patients with a history of cancer among patients hospitalized for COVID-19, and to identify factors associated with the risk of death and morbidity among patients with cancer.

## 2. Materials and Methods

### 2.1. Patients Selection

In order to estimate the excess risk of mortality and morbidity among patients with a history of cancer, we conducted a matched case–control study of patients aged >18 in two hospitals of the Lyon area in France, hospitalized for COVID-19 during the first epidemic wave (March–April 2020).

The “exposed patients” were all first consecutive patients with cancer that were hospitalized for COVID-19. A patient with cancer was defined as any patient who had a surgical procedure or a medical treatment for cancer in the past 5 years preceding the COVID-19 pandemic. The “control patients” were all of the patients without cancer. In both cohorts, a positive laboratory test (RT-PCR) proving SARS-CoV-2 infection was mandatory. COVID-19 severity was quantified by the WHO scale for clinical status [9]. Two “control patients” were matched to each individual cancer patient based on gender, age (<60 vs. 60–80 vs. >80), and comorbidities (chronic respiratory disease, congestive heart failure, chronic kidney disease, chronic liver disease, diabetes, and high blood pressure). Control patients were randomly selected among a list of potential matched controls for each patient with cancer. Cancer patients and matched controls were identified using the hospital information system. Individual patients’ data were manually collected and verified from patients’ electronic files. When patients tagged with cancer by the information system had no history of cancer after the manual quality control, the case was excluded from the study along with the two matched control patients. When control patients were found to have a history of cancer after the manual quality control, the control patient was excluded from the study, leaving the cancer patient with only one matched control.

The study was approved by local ethic board (N. HCL 20_158) according to French regulation and falls within the Reference Methodology N 4. of the French National Commission for data protection and freedom of information (CNIL) for which the HCL has signed commitment of compliance and respects of the General Data Protection Regulation (HCL register MR004_20_152).

### 2.2. Endpoints

The primary endpoint was the fatality rate within 60 days after the hospitalization for COVID-19. The secondary endpoints were the incidence of intubation and mechanical ventilation, length of hospital stay for COVID-19, length of intensive care unit (ICU) stay for COVID-19, and length of oxygen supplementation.

### 2.3. Additional Patients Characteristics

The description of COVID-19 severity at admission was performed using the World Health Organization (WHO) scale for clinical status [9]. Comorbidities, laboratory measures, and cancer history details were manually extracted from patients’ files by clinical research associates trained in the field of oncology. In our institutions, a clear statement of any limitation of therapeutic effort was expected in all patients’ electronic files based on patients’ clinical status, comorbidities, life expectancy, and preferences. These decisions, usually made on the day of patients’ admission to the hospital, were collected as a potential factor associated with early death due to COVID-19.

### 2.4. Statistical Analyses

Continuous variables were described using their median values, interquartile ranges (IQR), mean, and standard deviations. Binary variables were described using proportions.

The study planned to include *n* = 100 cancer patients in order to have a statistical power of >90% to be able to show a difference in the 60-days mortality rate if the mortality rate was 10% for control patients and 25% for cancer patients, given an alpha risk of 5% based on very early data at the onset of the COVID-19 pandemic.

A conditional logistic modeling of the odds ratio of early death associated with the history of cancer was performed. In order to identify the major characteristics associated with the increased risk of death among cancer patients, unadjusted logistic regression models were performed. Multivariable models were then created to investigate the relationship between variables significantly associated with the risk of early death in unadjusted models (*p* < 0.10). Only variables independently associated with the risk of early death were retained in the final model using a backward elimination (retention threshold *p* < 0.05). Survival probabilities over time were estimated using the Kaplan–Meier estimator for illustrative purposes.

For the analysis of the 3 outcomes captured by ordinal variables (number of days of hospitalization; number of days of hospitalization in ICU; and number of days with oxygen supplementation), the effect of cancer history was evaluated by a Poisson regression with adjustment of the matching factors.

Statistical analyses and illustrations were performed using R Software v3.5.0 (R Foundation for Statistical Computing, Vienna, Austria).

## 3. Results

### 3.1. Patients’ Characteristics

A total of 108 cancer patients hospitalized for COVID-19 between March and April 2020 were included and matched with 193 control patients. Median age was high, close to 76 years, and similar in both groups. The patients’ characteristics were well balanced between groups regarding gender, comorbidities, and age. The body mass index (BMI) was significantly lower among cancer patients (median BMI = 24 vs. 26 kg/m^2^ among non-cancer patients).

At admission for COVID-19, cancer and non-cancer patients had a similar description of the disease in terms of the type of symptoms and disease severity as assessed by the WHO scale for clinical status. Laboratory measures (C-reactive protein, troponin, lymphocytes, platelets) were not available for all included patients but were similar among cancer and non-cancer patients when available (Table 1).

In the cancer cohort (*n* = 108), the most frequent tumor types were hematologic malignancies (*n* = 28, 26%), breast cancer (*n* = 16, 15%), and lung cancer (*n* = 13, 12%). Forty-nine (45%) patients had a history of advanced or metastatic cancer. Within the 3 months before COVID-19, 21 patients (19%) had received cytotoxic chemotherapy, and 7 (5%) more had received a combination of rituximab and cytotoxic chemotherapy. Nine patients (8%) had received three or more lines of chemotherapy (Table 2).

### 3.2. Impact of Cancer History on COVID-19 Outcomes

The risk of early death (within 60 days) was higher among cancer patients (40% vs. 28%; OR = 2.0; 95% CI: 1.2–3.4; *p* = 0.0084) (Table 3, Figure 1). On the contrary, the risk of intubation and mechanical ventilation was lower among cancer patients (9% vs. 18%; OR = 0.31; 95% CI: 0.13–0.74; *p* = 0.0089). Cancer patients spent a few less days on average at hospital (18 vs. 19, adjusted relative rate = 0.94; 95% CI: 0.89–1.0; *p* = 0.041). The mean number of days spent in ICU was significantly lower among cancer patients (3 vs. 6, adjusted relative rate = 0.51; 95% CI: 0.45–0.58; *p* < 0.0001), as well as the mean number of days with oxygen (7 vs. 10, adjusted relative rate = 0.66; 95% CI: 0.60–0.72; *p* < 0.0001) (Table 3).

### 3.3. Factors Associated with Early Risk of Death among Cancer Patients

In the cohort of 108 patients with a history of cancer, the main factors associated with early death in univariate analysis were the severity of COVID-19 at admission, male gender, therapeutic limitation decided at admission for COVID-19, and number of previous chemotherapy lines ≥3. The cancer type, cancer status, and type of recent cancer therapy were not associated with risk of early death. In multivariate analyses, only the decision to limit the therapeutic intensity at admission and previous exposure to ≥3 lines of chemotherapy were independently associated with te risk of early death from COVID-19 (Table 4).

## 4. Discussion

The COVID-19 pandemic has dramatically impacted cancer patients for several reasons, including the difficulty for cancer patients to maintain full access to the healthcare system [10]. Very early at the start of the pandemic, cancer patients have been pointed out as a population at higher risk of infection by SARS-CoV-2 [2,11], and at higher risk of severe infection and death from COVID-19 [5,11,12,13].

Multiple cohorts of cancer patients with COVID-19 have already been reported, and within these cohorts, factors associated with fatalities were hematological cancer [5,13], lung cancer [12], being male [5,13], greater age [5,13], and other comorbidities [13,14].

All these cohorts are at high risk of bias because they mostly included hospitalized patients and the causal relationship between cancer or cancer treatments and COVID-19 severity could not be tested.

In this study, cancer patients hospitalized for COVID-19 were matched with non-cancer patients also hospitalized for COVID-19 according to known prognostic factors of the disease (gender, age, and number of comorbidities). The matching of the patients appeared to have been efficient, as the baseline clinical features of COVID-19 in cancer were very similar (Table 1) to those observed in non-cancer patients. Despite the well-balanced COVID-19 severity at admission, the 60-day death rate was significantly higher among cancer patients. This observation is unlikely to be linked to an intrinsic vulnerability to COVID-19 of cancer patients, as cancer patients spent on average less days with oxygen, less days in ICU, and were less likely to receive mechanical ventilation. We observed a higher risk of death among cancer patients in the latter stage of advanced disease (≥3 previous lines of chemotherapy), and among cancer patients with a decision to limit the intensity of the treatments at admission (Table 4). Therapeutic limitation is a plausible explanation of the overall higher risk of early death among cancer patients, and it can also explain the lower risk of receiving mechanical ventilation among cancer patients. The decision to limit the therapeutic intensity at admission was logically correlated with other factors determining this decision, such as the severity of COVID-19 at admission, the patient’s general status and comorbidities, and the prognosis of the underlying cancer, making the causal relationship of these factors with the risk of early death difficult to ascertain. Considering the higher risk of fatality among cancer patients, it seems unlikely that the lower morbidity observed among cancer patients (less days with oxygen, less days in ICU, and lower risk of receiving mechanical ventilation) is explained by a protective effect of cancer of the anticancer drugs against the excess immune response to COVID. On the contrary, the reduction of peripheral lymphocytes, including CD3+, CD4+, and CD8+ T cells, has been associated with COVID-19 severity and outcome [15,16]. The lung inflammatory process might be worse among cancer patients due to alveolar-capillary disruption caused by some chemotherapy drugs [17].

The observation that cancer and non-cancer patients have similar COVID-19 presentations but that cancer patients are more frequently limited in terms of treatment intensity is compatible with other previously published observations. In a report from New York City in which cancer patients were not limited in terms of therapeutic intensity (higher intubation rate compared with non-cancer patients), the rate of death was not significantly different between cancer and non-cancer patients [18]. In a more comprehensive cohort, including all ambulatory or hospitalized patients with COVID-19 and followed for cancer in a high-volume institution, the incidence and presentation of COVID-19 was similar in cancer patients as what was observed in the general population [19].

There are limitations to this study. First, the retrospective nature of this study and the relatively limited number of patients prevented us from assessing relationship between infrequent cancer characteristics and the risk of early death from COVID-19. However, the matching between cancer and non-cancer patients limited the risk of confusion bias when assessing the relationship between cancer and COVID-19 severity compared to previously published cohorts. The heterogeneity of the cancer cohort in terms of cancer type, treatment, and status limits the possibility of deriving a strong conclusion in any subgroup of cancer patients. Finally, the patients included in this study were all included during the first wave of COVID-19 in France while new variants might have a different relative impact on cancer patients and treatment options have evolved [20].

## 5. Conclusions

Despite these limitations, our study strongly suggests that COVID-19 has a similar presentation and severity in a general population of cancer patients compared with non-cancer patients. Acknowledging the detrimental collateral impact of cancer treatment disruption [21,22,23], these results suggest that cancer diagnostic, surgical, and medical treatment pathways should be maintained as normal as possible. Interpreting our results in light of the high immunogenicity of COVID-19 vaccination in most oncology patients, cancer patients should be encouraged to be vaccinated following the same rules as the general population [24,25], except the small population of severely immunosuppressed patients who need dedicated immunization strategies [25].

## Figures and Tables

**Figure 1 cancers-13-05283-f001:**
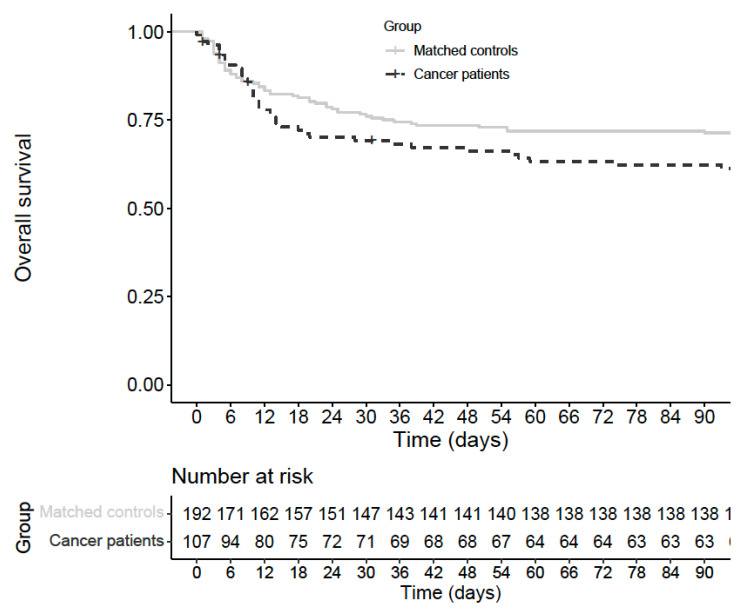
Overall survival from hospital admission among cancer patients and matched controls.

**Table 1 cancers-13-05283-t001:** Patients’ baseline characteristics.

Patient Characteristics	Cancer (*n* = 108)	Control (*n* = 193)	*p*
Age at treatment initiation, years, median (25th–75th) NA = 0	76 (65–73)	77 (67–85)	0.17
Age category			
<60	15 (14%)	22 (11%)	0.81
60–80	54 (50%)	98 (51%)	
>80	39 (36%)	73 (38%)	
Gender male (%) NA = 0	50 (46%)	93 (48%)	0.81
Median BMI at treatment initiation, years, median (25th–75th) NA = 40	24 (21–27)	26 (23–29)	0.006
Comorbidity			
Diabetes, NA = 6	27 (25%)	46 (24%)	0.89
Hypertension, NA = 10	55 (53%)	95 (51%)	0.81
Chronic respiratory disease, NA = 11	13 (12%)	26 (14%)	0.72
Chronic renal disease NA = 10	21 (19%)	36 (20%)	1
Cardiac disorder, NA = 24	16 (16%)	29 (16%)	1
Liver disease, NA = 8	3 (3%)	2 (1%)	0.36
Laboratory measures at admission			
C-reactive protein, NA = 29	82 (33–176)	70 (28–145)	0.2
Troponin, NA = 144	19 (10–43)	18 (10–55)	0.96
Lymphocytes, NA = 14	0.78 (0.50–1.3)	0.94 (0.64–1.4)	0.067
Platelets, NA = 17	200 (140–278)	215 (166–258)	0.44
ASAT, NA = 86	43 (30–55)	44 (33–71)	0.2
Symptoms at diagnosis (%)			
Cough, NA = 5	59 (56%)	121 (64%)	0.21
Fever, NA = 34	21 (22%)	42 (25%)	0.66
Respiratory distress, NA = 0	11 (10%)	20 (10%)	1
Gastrointestinal, NA = 9	27 (26%)	40 (21%)	0.46
Headache, NA = 8	4 (4%)	17 (9%)	0.1
Myalgia, NA = 9	10 (10%)	20 (11%)	0.84
Anosmia, NA = 6	7 (7%)	11 (6%)	0.8
Ordinal WHO scale for clinical status *			
4	50 (46%)	75 (39%)	0.19
5	45 (42%)	78 (40%)	
6	3 (3%)	5 (3%)	
7	10 (9%)	35 (18%)	
Medicines received, NA = 0			
Steroids	21 (19%)	23 (12%)	0.56
Hydroxychloroquin	6 (6%)	19 (10%)	0.28
Remdesivir	0 (0%)	1 (1%)	1
Any other therapy	10 (9%)	17 (9%)	1

* The patients’ clinical status was assessed on an ordinal scale as follows: 4, hospitalized, no oxygen therapy; 5, hospitalized, oxygen by mask or nasal prongs; 6, hospitalized, high flow oxygen; 7, Intubation and mechanical ventilation. NA: data not available.

**Table 2 cancers-13-05283-t002:** Description of the cancer characteristics in the cancer cohort.

Patient Characteristics	*N* = 108 (%)
Type of cancer, NA = 0	
Hematologic malignancy	28 (26%)
Lung cancer	13 (12%)
Breast cancer	16 (15%)
Prostate cancer	8 (7%)
Other solid cancer	43 (40%)
Stage, NA = 0	
Advanced or metastatic	49 (45%)
Localized, remission, observation only	59 (55%)
Lung or pleural metastasis	19 (18%)
Previous treatment in the last 3 months	
Chemotherapy	21 (19%)
Chemotherapy + rituximab	7 (6%)
Endocrine therapy	2 (2%)
Immunotherapy/Targeted therapy	7 (6%)
Cancer surgery	4 (4%)
Radiation therapy	4 (4%)
Previous exposure to chemotherapy	
No	67 (62%)
<3 lines	32 (30%)
≥3 lines	9 (8%)
Previous biological status before COVID diagnosis	
Lymphocyte (G/L), NA = 37	1.1 (0.7–1.7)
C-reactive protein (mg/L), NA = 59	19 (5–87)

**Table 3 cancers-13-05283-t003:** Patient outcomes in the cancer and matched control cohorts.

		*N* (%)	Conditional OR (95% CI)	*p*
Early death (within 60 days)	Cancer (*n* = 108)	43 (40%)	2.0 (1.2–3.4)	0.0084
	Matched control (*n* = 193)	54 (28%)	REF	
Intubation and mechanical ventilation	Cancer (*n* = 108)	10 (9%)	0.31 (0.13–0.74)	0.0089
	Matched control (*n* = 193)	35 (18%)	REF	
		median (25th–75th), mean (sd)	Adjusted relative rate (95% CI)	
Number of days with oxygen	Cancer (*n* = 108)	3 (0–10), 7 (10.2)	0.66 (0.60–0.72)	<0.001
	Matched control (*n* = 193)	6 (1–14), 10 (13.2)	REF	
Number of days in ICU	Cancer (*n* = 108)	0 (0–0), 3 (10.3)	0.51 (0.45–0.58)	<0.001
	Matched control (*n* = 193)	0 (0–3), 6 (14.1)	REF	
Number of hospital days	Cancer (*n* = 108)	11 (6–19), 18 (23)	0.94 (0.89–1.0)	0.041
	Matched control (*n* = 193)	12 (6–24), 19 (22)	REF	

**Table 4 cancers-13-05283-t004:** Univariate and multivariate analysis of risk of death according to patient and tumor characteristics within the cancer cohort.

Binomial Logistic Regression Models for Death						
			Unadjusted Analysis		Adjusted Analysis (Backward Procedure)	
	*N* (%)	Early Deaths within 60 Days *N* (%)	OR (95% CI)	*p*	OR (95% CI)	*p*
Age category NA = 0						
<60	15 (14%)	5 (33%)	REF	0.79	NI	NI
60–80	54 (50%)	23 (43%)	1.48 (0.45–4.9)			
>80	39 (36%)	15 (38%)	1.25 (0.36–4.4)			
Gender NA = 0						
Female	58 (54%)	18 (31%)	REF	44	NI	NI
Male	50 (46%)	25 (50%)	2.2 (1.0–4.9)			
Number of comorbidities						
0	35 (32%)	10 (29%)	REF	0.24	NI	NI
1–2	59 (55%)	27 (46%)	2.1 (0.86–5.2)			
≥3	14 (13%)	6 (43%)	1.9 (0.52–6.8)			
Setting, NA = 0						
Advanced or metastatic	49 (45%)	24 (49%)	REF	0.076	NI	NI
Localized, remission, observation only	59 (55%)	19 (32%)	0.49 (0.23–1.1)			
Hematologic malignancy						
No	80	32 (40%)	REF	0.95	NI	NI
Yes	28	11 (39%)	0.97 (0.40–2.3)			
Baseline BMI, NA = 7						
<18	8 (8%)	3 (38%)	1.0 (0.22–4.4)	0.6	NI	NI
18–30	85 (84%)	32 (38%)	REF			
>30	8 (8%)	5 (62%)	2.8 (0.62–12.3)			
Lung or pleural metastasis, NA = 0						
No	89 (82%)	32 (36%)	REF	0.079	NI	NI
Yes	19 (18%)	11 (58%)	2.4 (0.89–6.7)			
Previous exposure to chemotherapy						
None	67 (62%)	24 (36%)	REF	0.052	REF	0.012
<3 lines	32 (30%)	12 (38%)	1.1 (0.45–2.6)		2.8 (0.84–9.5)	
≥3 lines	9 (8%)	7 (78%)	6.3 (1.2–32.6)		13.6 (1.8–100)	
Previous chemotherapy treatment in the last 3 months						
No	80 (74%)	30 (38%)	REF	0.41	NI	NI
Yes	28 (26%)	13 (46%)	1.4 (0.61–3.4)			
Previous rituximab treatment in the last 3 months						
No	101 (94%)	40 (40%)	REF	0.87	NI	NI
Yes	7 (6%)	3 (43%)	1.1 (0.24–5.4)			
Therapeutic limitation decided at the admission						
No	65 (60%)	11 (17%)	REF	<0.001	REF	<0.001
Yes	43 (40%)	32 (74%)	14.3 (5.6–36.7)		15.3 (5.2–44.7)	
Ordinal scale for clinical status *						
4	50 (46%)	13 (26%)	REF	0.016	REF	0.11
5	45 (42%)	22 (49%)	2.7 (1.2–6.4)		2.6 (0.83–7.8)	
≥6	13 (12%)	8 (61%)	4.6 (1.3–16.4)		4.7 (0.91–24.6)	

* The patients’ clinical status was assessed on an ordinal scale as follows: 4, hospitalized, no oxygen therapy; 5, hospitalized, oxygen by mask or nasal prongs; 6, hospitalized, high flow oxygen; 7, Intubation and mechanical ventilation.

## Data Availability

The data presented in this study are available on request from the corresponding author. The data are not publicly available due to the necessity to make every effort to maintain patients non identifiable according to European law.

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
