# Peer review of "COVID-19 Presentation and Outcomes among Cancer Patients: A Matched Case-Control Study"

_cancers, 2021, doi:10.3390/cancers13215283_

Round 1

Reviewer 1 Report

1) It would be helpful if the authors were able to show the difference in expression of genes and are associated with cancer patients compared to Healthy individuals.  

2) The authors should provide the list of genes in cancer patient and their effects with relation to SARS-CoV-2  infection and mortality. Everybody knows that in cancer patients the immune system is weak and the infection progression is high compared to healthy Individuals. 

Author Response

We agree with the reviewer, but not genomic or transcriptomic analyses were performed in routine during the management of these patients, and such analysis is then not available.

Reviewer 2 Report

The study represents a retrospective analysis of >18 yrs patients hospitalized for COVID-19 at Lyon area in France. This study is well organized and nicely represented. This study could be accepted after a few modifications.

  1. Clearly explain how the study answers the clinical questions.
  2. How was the sample size calculated? What is the power of study?
  3. How were the patients have chosen, random or consecutive?
  4. The discussion section is very limited. Authors should deliberate on different aspects of their study. There is a need to discuss the current status of treatment. I also suggest including in the discussion PMID: 33134842.
  5. Please provide more detail about the matching method in the case-control study, 
  6. The study included only two hospitals in Lyon, France Therefore, the Title needs to be revised to better reflect the content of the article.
  7. Insert in the discussion about the scientific impact of the study.

Author Response

Reviewer 2:

The study represents a retrospective analysis of >18 yrs patients hospitalized for COVID-19 at Lyon area in France. This study is well organized and nicely represented. This study could be accepted after a few modifications.

  1. Clearly explain how the study answers the clinical questions.

The manuscript has been edited accordingly line 72 “In order to estimate the excess risk of mortality and moribidity among patients with a history of cancer, we conducted a matched case–control study of patients aged >18 in two hospitals of the Lyon area in France, hospitalized for COVID-19 during the first epidemic wave (March-April 2020). “

  1. How was the sample size calculated? What is the power of study?

Line 115: “The study planned to include n=100 cancer patients in order to have a statistical power of >90% to be able to show a difference in the 60-days mortality rate if the mortality rate was 10% for control patients and 25% for cancer patients, given an alpha risk of 5% based on very early data at the onset of the COVID-19 pandemic. “

  1. How were the patients have chosen, random or consecutive?

Cancer patients were consecutive. Control patients were matched to cancer patients, so randomly chosen. The methods section has been edited to clarify this line.

  1. The discussion section is very limited. Authors should deliberate on different aspects of their study. There is a need to discuss the current status of treatment. I also suggest including in the discussion PMID: 33134842.

The discussion section has been extended according to the reviewer comment, even if we want to keep it as simple as possible. A lot of papers have been produced during this COVID pandemic, and it might not be helpful to discuss COVID-related topics not directly related to our data. The reference suggested by the reviewer has been included in the introduction section, not in the discussion section.

  1. Please provide more detail about the matching method in the case-control study, 

The procedure was simple but is now explained more extensively line 85.

  1. The study included only two hospitals in Lyon, France Therefore, the Title needs to be revised to better reflect the content of the article.

This is true and clearly reported in the methods section line 74. To our understanding, we see not contradiction between the title and this fact. Could you please clarify if you really perceive a contradiction so we can edit accordingly?

  1. Insert in the discussion about the scientific impact of the study.

The observation that COVID-19 has a similar presentation and severity in a general population of cancer patients compared with non-cancer patients is a very strong scientific statement, that is really useful for both scientific research and also clinical practice. It is supported by other published articles, but contradicts priori belief. This is the main scientific impact of the study.

Reviewer 3 Report

I have read with interest the paper by Julien Péron entitled “COVID-19 presentation and outcomes among cancer patients, a matched case-control study”. I have some comments to be addressed to the authors.

What is difficult to understand, at least for me, is why the cancer patients showed reduced these outcomes: risk of intubation, number of days with oxygen, the length of stay in ICU, and the length of hospital stay. Should all these outcomes be explained by both an increased early death of these patients and the therapeutic limitation?  Apart those patients in a terminal state of the cancer disease, who are obviously expected to go towards an early death, could the reduced outcomes, listed above, be interpreted in a different way? In other words, could the immune system of cancer patients be more prone to counteract the CoV 19 infection? If we consider that a poor prognosis of the CoV 19 disease is almost due to an exaggerated immune-coagulative reaction to the virus attack, in the cancer patients the immune system, perhaps, is less reactive and so much more tolerant in face of the virus invasion. I would like to read the authors’ response to this hypothesis. In any case the authors should better explain the results of their study. 

Author Response

I have read with interest the paper by Julien Péron entitled “COVID-19 presentation and outcomes among cancer patients, a matched case-control study”. I have some comments to be addressed to the authors.

What is difficult to understand, at least for me, is why the cancer patients showed reduced these outcomes: risk of intubation, number of days with oxygen, the length of stay in ICU, and the length of hospital stay. Should all these outcomes be explained by both an increased early death of these patients and the therapeutic limitation?  Apart those patients in a terminal state of the cancer disease, who are obviously expected to go towards an early death, could the reduced outcomes, listed above, be interpreted in a different way? In other words, could the immune system of cancer patients be more prone to counteract the CoV 19 infection? If we consider that a poor prognosis of the CoV 19 disease is almost due to an exaggerated immune-coagulative reaction to the virus attack, in the cancer patients the immune system, perhaps, is less reactive and so much more tolerant in face of the virus invasion. I would like to read the authors’ response to this hypothesis. In any case the authors should better explain the results of their study. 

We really understand the comment of the reviewer, as it might be seen as contra intuitive that cancer patients have a higher risk of death from COVID but a lower risk of intubation, intensive care requirement etc… The hypothesis suggested by the reviewer that anticancer treatments, immunosuppressive for lots of them, could have a protective effect against the excessive immune response to COVID is appealing, but is challenged by several observations. We included this in the discussion section: “Considering the higher risk of fatality among cancer patients, it seems unlikely that the lower morbidity observed among cancer patients (less days with oxygen, less days in ICU, and lower risk to receive mechanical ventilation) is explained by a protective effect of the cancer of of the anticancer drugs against the excess immune response to COVID. On the contrary the reduction of peripheral lymphocytes, including CD3+, CD4+, and CD8+ T cells, has been associated with COVID-19 severity and outcome [15,16]. The lung inflammatory process might be worse among cancer patients due to alveolar-capillary disruption caused by some chemotherapy drugs [17]. “

The main hypothesize that we try to highlight through the discussion section is that cancer and non-cancer patients have similar COVID-19 presentations and severity, but that cancer patients are more frequently limited in term of treatment intensity and are then at higher risk of early death to COVID. The discussion has been rephrased ‘line224 to 228’ to clarify our interpretation of the results.